# Retinoid-X receptor agonists increase thyroid hormone competence in lower jaw remodeling of pre-metamorphic *Xenopus laevis* tadpoles

**Brenda J. Mengeling**📷*, **Lara F. Vetter**📷, **J. David Furlow**

Department of Neurobiology, Physiology, and Behavior, College of Biological Sciences, University of California, Davis, California, United States of America

* mengeling@ucdavis.edu

## Abstract

Thyroid hormone (TH) signaling plays critical roles during vertebrate development, including regulation of skeletal and cartilage growth. TH acts through its receptors (TRs), nuclear hormone receptors (NRs) that heterodimerize with Retinoid-X receptors (RXRs), to regulate gene expression. A defining difference between NR signaling during development compared to in adult tissues, is competence, the ability of the organism to respond to an endocrine signal. Amphibian metamorphosis, especially in *Xenopus laevis*, the African clawed frog, is a well-established in vivo model for studying the mechanisms of TH action during development. Previously, we've used one-week post-fertilization *X. laevis* tadpoles, which are only partially competent to TH, to show that in the tail, which is naturally refractive to exogenous T3 at this stage, RXR agonists increase TH competence, and that RXR antagonism inhibits the TH response. Here, we focused on the jaw that undergoes dramatic TH-mediated remodeling during metamorphosis in order to support new feeding and breathing styles. We used a battery of approaches in one-week-old tadpoles, including quantitative morphology, differential gene expression and whole mount cell proliferation assays, to show that both pharmacologic (bexarotene) and environmental (tributyltin) RXR agonists potentiated TH-induced responses but were inactive in the absence of TH; and the RXR antagonist UVI 3003 inhibited TH action. Bex and TBT significantly potentiated cellular proliferation and the TH induction of *runx2*, a transcription factor critical for developing cartilage and bone. Prominent targets of RXR-mediated TH potentiation were members of the matrix metalloprotease family, suggesting that RXR potentiation may emphasize pathways responsible for rapid changes during development.

## Introduction

An organism's acquired ability to respond both qualitatively and quantitatively to a physiological signal, defined as competence, is distinguished between endocrine signaling during

**Data Availability Statement:** All relevant data are within the paper and its Supporting Information files.

**Funding:** JDF and BJM: R21 ES026271, National Institues of Environmental Health Sciences, https://www.niehs.nih.gov/ JDF: P42 ES004699, National Institute of Environmental Health Sciences/ Superfund Research Program, https://www.niehs.nih.gov/ The funders had no role in study design, data collection and analysis, decision to publish, or preparation of the manuscript.

**Competing interests:** The authors have declared that no competing interests exist.

development, which tends to lead to irreversible, organizational effects, from that of healthy adult tissues, which controls the functioning of tissues and organs to maintain homeostasis. Thyroid hormone (TH) action regulates many aspects of vertebrate development including cartilage growth and skeletogenesis [1–4]. Over developmental time, the vertebrate organism traverses from low to high TH competence [5]. Vertebrate development depends upon appropriate timing and concentrations of TH for good biological outcomes. During human development, adverse outcomes arise from both insufficient and excessive TH [1, 6–10]. However, analysis of the effects of TH on mammalian development are confounded by maternal effects due to the nature of intrauterine growth. Amphibian metamorphosis, the process through which free-living larval tadpoles develop into adult frogs provides an accessible and dramatic model for direct investigation of the role TH plays during vertebrate development [11–14]. Metamorphosis is initiated and maintained through the action of TH [15–18]. The African clawed frog, *Xenopus laevis*, is an effective laboratory model for assessing the role of TH throughout development, and its metamorphosis has been shown to model the essential perinatal surge in TH signaling in humans [11, 14].

In all vertebrates, TH acts through the thyroid hormone receptors (TRs), which are DNA-binding, ligand-regulated transcription factors of the nuclear receptor (NR) superfamily [19, 20]. THs are identical across all taxa, and the TRs are highly conserved between *X. laevis* and humans [12, 13]. Two isoforms of TR are expressed from two different genes, TRα and TRβ. In *X. laevis* tadpoles, TRα is expressed before synthesis of THs commences [21], whereas TRβ expression is induced after the nascent thyroid gland begins to synthesize THs through TH binding to TRα; it is a direct target gene of TRs [22, 23]. 3,3',5-triiodo-L-thyronine (T3) is the TH with the highest affinity for the TRs [24–26].

TRs heterodimerize with another NR, the retinoid-X receptors (RXRs) [27]. RXRs bind several natural ligands, including 9-cis retinoic acid, and they can dimerize with many different NRs in addition to TRs [28]. The TR-RXR heterodimer shows higher affinity for DNA, especially in the presence of T3, than the TR-TR homodimer [29]. In most adult tissues and cells, RXR ligands are unable to affect the action of the TR-RXR heterodimer [30, 31]. Pituitary cells are an exception, wherein RXR ligands do affect the ability of TR to control the hypothalamus-pituitary-thyroid (HPT) axis [32]; the biological reasons for this are not understood. In fact, the pharmaceutical RXR agonist used in this study, bexarotene (brand name Targretin, Bex), produces severe hypothyroidism in patients given the drug, which limits its use as a chemotherapeutic [33–35]. However, given the inability of RXR ligands to affect TR function in peripheral tissues, such as the liver—a major site of TR function—the TR-RXR heterodimer is generally considered to be an example of a "non-permissive" RXR heterodimer, meaning that only the ligand for the TR, T3, can induce activation.

Due to the importance of TH signaling for proper development, man-made chemicals that disrupt TR action have the potential to produce adverse outcomes [36]. Tributyltin (TBT) is a pervasive environmental pollutant from its use as an antifoulant in marine paints that was the first described endocrine disruptor, when it was discovered that exposure to TBT caused marine gastropods to develop imposex phenotypes, where female gastropods develop male secondary sex characteristics [37, 38]. Mechanistic work determined that TBT functioned through the mollusk RXR, and that treating marine gastropods with either 9-cis retinoic acid or TBT produced the same imposex phenotype [39–41]. In our rat pituitary reporter cell line, TBT behaved like Bex, strongly suggesting that it was functioning as an RXR agonist [42, 43]. These results left open the question as to whether RXR agonists in a developing organism would behave like RXR agonists in most adult tissues (i.e. RXR agonists have no effect on TR action) or in our pituitary reporter cell line (i.e. RXR agonists could modulate TR function).

In order to understand the role of disruptors of TR and RXR on developmental TH action, we developed a suite of quantitative assays to assess function and possible disruption of TH action in 1-week post-fertilization (1wk-PF) tadpoles (NF 48) [44]. 1wk-PF tadpoles express TRα, but they do not yet have an active thyroid gland; therefore, they are TH negative and are considered pre-competent [21]. Addition of T3 to their rearing water activates many metamorphic pathways, but the addition of T3 does not make their TH competence complete. For example, tail resorption, the last step of metamorphosis, is minimal even under supraphysiological doses of exogenous T3 [45]. We found that co-treatment of Bex or TBT with T3 significantly potentiated the action of T3 in the tail [43, 46]. In effect TBT/Bex increased T3 competence in the tail to near metamorphic levels. At the transcriptomic level, we found that TBT acted identically to Bex, solidifying that the mechanism of TBT action on TH function was at the level of RXR agonism [46].

Amphibian metamorphosis affects almost every tissue system and cell fate decision. Some larval tissues are resorbed like tail and gills, some adult tissues are formed de novo like limbs, and other larval tissues are remodeled like the lower jaw (LJ). Although both retinoic acids (RAs) and TH are important for appropriate craniofacial and jaw development, with RAs acting through retinoic acid receptors (RARs), another NR family member that heterodimerizes with RXRs, they function at different times. RAR signaling is necessary early for neural crest cell specification and migration [47–49]. TH action is required for correct maturation and ossification [1], which occurs late in development. In tadpoles these processes are separated by several weeks, with jaw cartilages having been formed by 98 h-PF (NF-45) [50], and ossification of the jaw occurring late in metamorphosis (approximately two months PF) [51, 52]. At the time of our assaying the effects of RXR ligands on TH action, 1-wk-PF (NF-48), the tadpoles have entered a multi-week period of isometric growth [51], wherein further morphological changes do not occur without exogenous intervention. During metamorphosis, TH induces the jaw to remodel to facilitate the transition from an herbivorous tadpole to a carnivorous adult frog. Visible jaw morphological changes start at NF 59, which is approximately 45 days post-fertilization (PF) under ideal rearing conditions [52, 53]. Thomson describes three phases of Meckel's cartilage (MC) development in the LJ: 1) a lag phase (NF 57–59) with low levels of cell proliferation, 2) a division phase (NF 60–62) of rapid cell division, and 3) a synthesis phase (NF 62–66) wherein the matrix content of the cartilage increases significantly [54, 55]. Rose showed that tadpoles prior to NF 57 (~41 days PF) respond to the TH but the beak-like morphological changes that result are not seen in a natural metamorphosis [53]. Between NF 48 and NF 57 significant, non-TH-induced isometric growth occurs to the cartilages of the lower jaw, and this growth appears to be essential for producing appropriate morphology upon TH administration. Bearing this in mind, we investigated whether RXR ligands were able to potentiate the T3-induced changes that are possible at NF 48, where we have an extant suite of quantitative assays to monitor potential disruption of T3 action, and when the jaw is developmentally quiet [44]. We found that both Bex and TBT potentiated T3-induced proliferation, the activation of *runx2*, a transcription factor necessary for maturation of cartilage and bone ossification, and the matrix metalloproteases *mmp11* and *mmp13l*. In addition, the RXR antagonist UVI 3003 (UVI) [56] prevented T3-induced morphological changes, inhibited proliferation, and it selectively inhibited gene transcription. Finally, Bex and TBT still potentiated T3 action in the LJ in tadpoles at NF 54, which are considered prometamorphic and fully competent to respond to THs.

## Materials and methods

### Reagents

3,3',5-triiodo-L-thyronine (T3, T6397-100MG) and tributyltin chloride (TBT, T50202-5G) were purchased from MilleporeSigma (Burlington, MA) and Bexarotene (Bex, 5819/10 and

UVI 3003 (UVI, 3303/10) were purchased from Tocris Biosciences (Bio-Techne, Minneapolis, MN). All treatment ligands were dissolved or diluted in dimethyl sulfoxide (DMSO, Thermo Fisher Scientific, Waltham, MA). oLH (ovine luteinizing hormone) was purchased through the National Hormone and Peptide Program (Los Angeles, CA), pregnant mare serum gonadotropin was purchased from Thermo Fisher Scientific, and tricaine methanesulfonate was purchased from Western Medical Supply (Arcadia, CA).

## Animal husbandry

The laboratory has an approved University of California Davis Institutional Animal Care and Use protocol that covers the husbandry and mating of adult *Xenopus laevis* frogs and ligand exposure of larval tadpoles. Wild-type *X. laevis* frogs were mated and embryos cultured as described [44].

## Tadpole precocious metamorphosis morphology assay

NF 48 (1-week post-fertilization) tadpoles were treated, fixed for photography, and dorsal head photos taken using a Leica DFC3000 G camera on a Leica MZLFIII microscope as described [43, 44]. Treatment concentrations, unless otherwise indicated, were 10 nM T3, 30 nM Bex, 1 nM TBT, and 600 nM UVI, based upon previous results. The angle of the lower jaw was measured using the FIJI [57] distribution of ImageJ [58]. GraphPad Prism 9 (GraphPad Software, La Jolla, CA) was used to generate box and whisker plots, where boxes represent the $25^{th}$ to $75^{th}$ percentiles with the bar at the median, and whiskers are maximum and minimum values. For statistical analyses, each animal counted as an individual, and 2 clutches (ten tadpoles/clutch) were assayed independently to control for clutch-to-clutch variability. NF 54 tadpoles were treated as NF 48 animals except that the volume/tadpole of rearing water was increased to 50 ml, and treatments were stopped at three days rather than five, due to the extreme gill resorption in the T3 + Bex animals. Three independent clutches of NF 54 tadpoles were used with 4–5 tadpoles per clutch.

## Alcian blue staining of cartilage

NF-48 tadpoles treated and fixed as for morphology, were stained with Alcian blue to visualize cartilage based on guidelines for whole mount Xenopus [59]. In brief, fixed tadpoles were bleached overnight in 50% ethanol/3% hydrogen peroxide and then transferred through an ethanol series up to 100% ethanol. Dehydrated tadpoles were stained for 2–3 h in 0.2% Alcian blue 8GX (Sigma-Aldrich, A5268)/70% ethanol/30% glacial acetic acid. Samples were washed 2–3 times in 70% ethanol/30% glacial acetic acid, and then allowed to destain over night. Tadpoles were rehydrated through a decreasing ethanol series, then cleared in 0.25% trypsin/30% saturated borax for 30 min at room temperature, and then stored at 4°C in PBS/0.02% sodium azide. Alcian blue stained tadpoles were photographed using a Leica MZ16F microscope and a Leica DFC 500 camera.

## Immunohistochemistry of lower jaws for proliferation

The lower jaws from tadpoles fixed as for morphology were removed as follows: a straight cut was made just posterior to the olfactory epithelium and anterior to the eyes. The upper and lower jaw were separated, and two diagonal cuts were made on the outer rim of the jaw to separate the cartilage from the excess tissue. LJs were treated as described for immunohistochemical analysis of phospho-Histone H3 reactivity [44, 60]. Anti-phospho-Histone H3 (Ser10) was from EMD Millepore (06–570, 1/300 dilution), and goat anti-rabbit IgG (H+L) conjugated

with Alexa Fluor 488 was from Molecular Probes (A11008, 1/400 dilution). Positive cells were counted from blinded images using the Cell Counter tool of Fiji and normalized to the area counted to control for changes in jaw size with treatment.

### Gene expression

Tadpoles were treated with ligands for 48 hours as for morphology and as described [43, 46], using a 2-way ANOVA design: vehicle (DMSO), T3, RXR ligand, and T3 + RXR ligand. Lower jaws were isolated from unfixed tadpoles as for immunohistochemistry. Pools of 15 LJs from a single clutch were used for total RNA extraction. LJ tissue was disrupted and homogenized by bead beating with two 0.125-inch stainless steel beads for 1 minute in a Mini-Beadbeater-16 (Biospec Products, Bartlesville, OK). Total RNA was extracted using the RNeasy Plus Mini Kit per the manufacturer's instructions (Qiagen, Germantown, MD). Total RNA was quantified using a NanoDrop (Thermo Fisher Scientific, Waltham, MA). One microgram of total RNA was used to synthesize cDNA with the High-Capacity Reverse Transcription Kit (Thermo Fisher Scientific), and 0.5 µl of cDNA from a 20-µl reaction was used in a 10-µl reaction using PowerUp SYBR Green Master Mix (Thermo Fisher Scientific) in a Roche LightCycler 480. The *X. laevis rpl8* gene was used as a normalizer. Statistics were performed using 2-way ANOVA analysis with a Sidak's multiple comparison test (MCT) in GraphPad Prism 9. Sequences for the primers used for quantitative PCR are given in Table 1.

### Transgenic tadpole luciferase reporter assay

NF 54 tadpoles, sorted at 1wk-PF for GFP+ expression in the eye lens, were staged by assessing morphology of the hind limb, according to the normal scale by Nieuwkoop and Faber [61], and then treated through their rearing water for two days as previously described [46]. Treatment concentrations were 10 nM T3 and 2 nM TBT. No mortality arose from the treatments over the treatment period. After treatment, tadpoles were anesthetized in 0.1% MS-222 (Western Medical) buffered with 0.1% sodium bicarbonate. The LJs were excised and minced on ice prior to freezing and then processed and assayed as described [44]. Each animal was treated as an individual for statistical purposes (n = 9 per treatment) from two independent clutches (4 animals in one clutch and 5 in the other). Two-way ANOVA using clutch and treatment as covariates with Tukey's MCT to compare treatments was used from GraphPad Prism 9.

## Results

### RXR agonists potentiate T3-induced morphological changes to the lower jaw, and an RXR antagonist abrogates T3 effects

Using our precocious metamorphosis assay system, we treated *X. laevis* 1wk-PF tadpoles (NF 48) for five days by exposure through their rearing water with vehicle or 10 nM T3 in the

**Table 1.**

| Gene | Forward | Reverse | Product Size (bp) | Allele specificity |
|---|---|---|---|---|
| aurkb | AACATGGCCGCTTTGATGAG | ATGTCTCTGTGGATGACTTTCCTTTC | 100 | both |
| mmp11.L | GCCGGCTCATGTTTCTTACC | GCAGTTCTACTGATCCCATTGC | 129 | L |
| mmp13l | GAAGAAGCCAGGACCTTGGAT | CAAATTGCAGAGCTCCGTTGA | 133 | both |
| rpl8 | GGTTGCATTCCGTGATCCTTA | GCTGAGCTTTCTTGCCACAGT | 107 | both |
| runx2 | GCCGCAACTTGCCTTATGTC | GGCTCAAAGGCAAGCGATTT | 112 | S |
| thibz | TGAACCTCAACCAATGCCTCA | CCAGAAGCACCTCCCTTAAACC | 150 | both |
| thrb | GTGCCAGGAAGGTTTCCTCTT | GGTCGGTGACTTTCATCAGCA | 100 | both |

presence or absence of RXR ligands. This treatment period did not result in animal mortality under any of the treatment conditions. Previously, we found that 30 nM Bex and 1 nM TBT produced maximal, non-toxic responses, and so we used them here [43, 46]. The dorsal head photos in Fig 1 show representative animals from each treatment regimen. Vehicle treatment resulted in normal tadpole morphology (Fig 1A), and treatment with the RXR ligands in the absence of T3 (Fig 1B–1D) did not result in morphological changes. Treatment with 10 nM T3 (Fig 1E) resulted in visible gill resorption and decreased the angle of the LJ. Co-treatment with either RXR agonist, Bex or TBT, potentiated the T3-inductions of gill resorption and the angle of Meckel's-infrarostral (IR) cartilages of the LJ (Fig 1F and 1G). However, co-treatment with the RXR antagonist UVI 3003, abrogated the effect of T3 on both morphologic phenotypes.

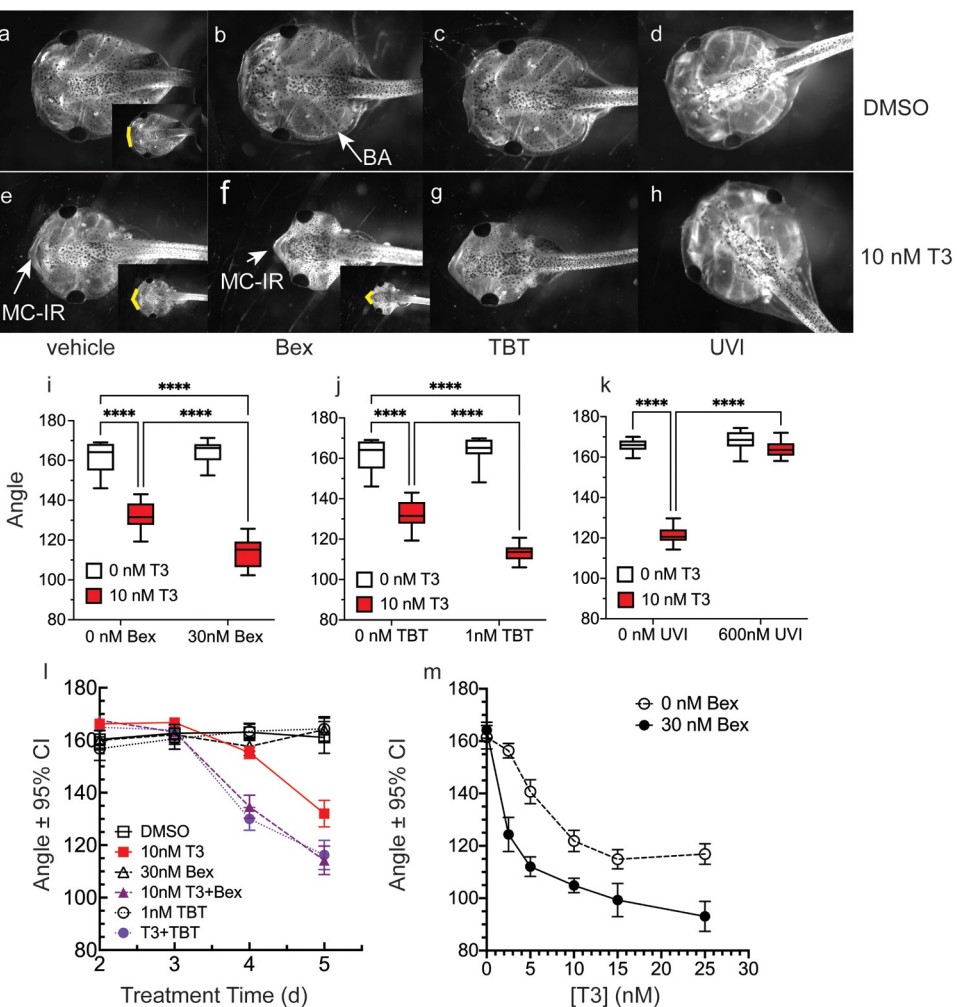

**Fig 1. RXR agonists potentiate T3-induced changes to lower jaw morphology, while an RXR antagonist abrogates T3 action.** a-h: Representative dorsal head photos of tadpoles treated for five days starting at 1wk-PF (as indicated: DMSO, 10 nM T3, and 30 nM Bex, 1 nM TBT, 600 nM UVI with or without T3). i-k: Quantification of changes to the LJ angle. Boxes represent 25th-75th percentiles with the line at the median (n = 10–15 from 2–3 clutches), and whiskers represent the min and max values. Statistics show results from Sidak's multiple comparison test in conjunction with 2-way ANOVA (****, p < 0.0001). l: Effect of Bex and TBT on T3-induced LJ angle changes as a function of time. Data points represent means from 20 animals from two different clutches; error bars delineate the 95% confidence intervals, indicating statistical significance. m: Treatment with 30 nM Bex augments LJ angle narrowing as a function of T3 dose. Statistics are the same as in the time course, although the clutches were different.

In order to quantify the effects of T3 and the RXR ligands on Meckel's and IR cartilages, we measured the angle of the LJ (Fig 1I–1K) from independent clutches of tadpoles, using ten animals per clutch. The inset photos (Fig 1A, 1E and 1F) show the change in angle that was measured. This facile and highly reproducible measurement, which is very useful for screening purposes, was validated through cartilage staining (see below). Protrusion of the Meckel's and IR cartilages caused a decrease in the LJ angle. Fig 1I shows that in the presence of T3, 30 nM Bex significantly potentiated the decrease in the LJ angle (compare red boxes). 2-way ANOVA analysis indicated significance for the interaction between T3 and Bex ($p < 0.0001$). As with our study on the effects of RXR agonists on T3-induced tail resorption, 1 nM TBT behaved almost identically to 30 nM Bex; the interaction between T3 and TBT was significant ($p < 0.0001$). In contrast, co-treatment of T3 and the RXR antagonist UVI prevented T3 action, and the LJ angle was not significantly changed from vehicle-treated tadpoles (Fig 1K); however, due to the strong abrogation of the T3-induction by UVI, the interaction between T3 and T3+UVI was still significant by 2-way ANOVA ($p < 0.0001$). Fig 1L shows the LJ angle measurement as a function of treatment time. Again, co-treatment of either Bex or TBT with T3 caused an identical response that showed an acceleration of the LJ cartilage protrusion. Tadpoles treated for four days with T3 plus RXR agonist had the same decrease in LJ angle as tadpoles treated for five days with T3-alone. Over a T3-dose curve (Fig 1M), the T3-induced decrease in LJ angle was significant starting at 5 nM T3 (error bars represent the 95% confidence interval), and all doses of T3 in the presence of Bex showed a significantly reduced LJ angle compared to T3-alone, such that 5 nM T3 plus Bex/TBT produced the same LJ angle as 15 nM T3, which is the dose that produces the maximal change in LJ angle.

The angle of the LJ measurement provides a straightforward method to determine whether a treatment compound can disrupt TH signaling in the LJ; however, it doesn't allow us to make statements at the level of the cartilages that provide the structure of the LJ. Therefore, we stained the cartilage with Alcian blue in 1wk-PF, whole tadpoles treated for five days with T3 in the presence or absence of Bex and TBT. Cartilage staining in tadpoles treated with only Bex or TBT was indistinguishable from vehicle-treated (DMSO) tadpoles. Fig 2A–2D shows representative ventral-side heads for each treatment stained with Alcian blue to visualize the cranial cartilages. In the vehicle-treated (DMSO, Fig 2A) the four cartilages visible from the ventral side are all present and distinct: the IR, Meckel's (MC), the ceratohyal (CH), and the branchial arches (BA) of the gills. The IR and MC appear as two distinct cartilages. After treatment with T3 (Fig 2B), the BA are partially resorbed, and the IR and MC appear fused. Co-treatment of either Bex (Fig 2C) or TBT (Fig 2D) potentiates T3 action to the extent that BA resorption appears complete, while the LJ cartilages remain deeply stained. We found that T3-alone or in combination with either Bex (Fig 2E) or TBT (Fig 2F) did not result in an increase in the length of the MC; this agrees with a previous report of exogenous TH induction of metamorphosis [53]. In addition, measurement of the LJ angle from the stained MC and IR (Bex, Fig 2G and TBT, Fig 2H) gave results nearly identical to those from the unstained DH photo (Fig 1I and 1J), validating the use of the more facile measurement of fixed, unstained tadpole LJ angles.

## RXR agonists potentiated, and the antagonist inhibited, T3-induced cellular proliferation in Meckel's and IR cartilage

In young tadpoles, exogenous T3 administration triggers cell proliferation in several tissues, including the LJ [60]. We excised LJs after four days of treatment for whole mount immunohistochemistry (IHC) of the mitotic marker phospho-Ser10 Histone 3 (pH3) to assess the effects of T3 and RXR ligands on cellular proliferation in Meckel's-IR cartilage. 1wk-PF tadpoles were

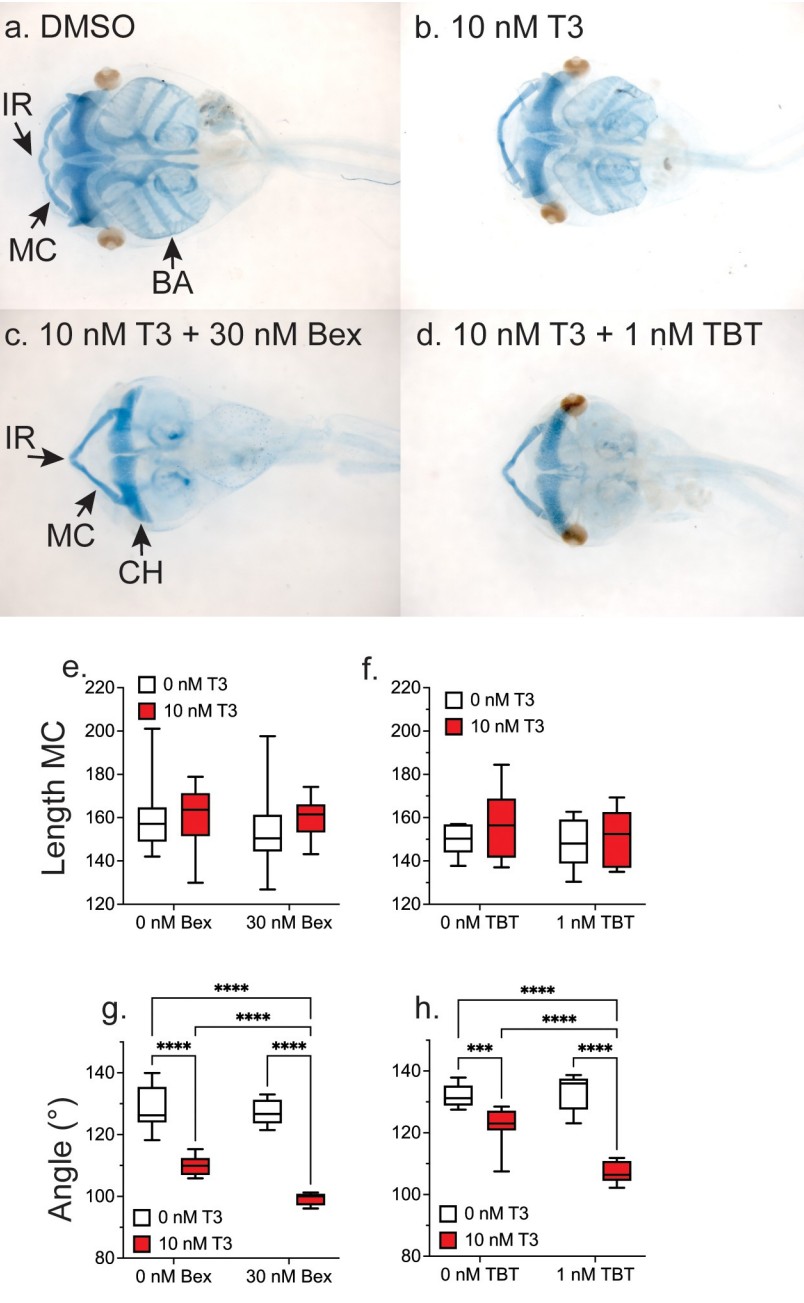

**Fig 2. RXR agonists potentiate T3-induced remodeling of LJ cartilage and resorption of gill cartilage.**
Representative ventral head photos of tadpoles treated for five days starting at 1wk-PF with vehicle (DMSO, a), 10 nM T3 (b), 10 nM T3 + 30 nM Bex (c), or 10 nM T3 + 1 nM TBT (d) and then stained with Alcian blue to visualize cartilage. MC, Meckel's cartilage; IR, infracostal cartilage; CH, ceratohyal cartilage; and BA, branchial arches. e-f: Meckel's cartilage lengths do not change with treatment. g-h: Changes in LJ angle measured from the end of the MC to the middle of the IR to the end of the opposite MC. Box plots are as in Fig 1 (n = 10 from 2 clutches of five per treatment. Statistics show results from Sidak's multiple comparison test in conjunction with 2-way ANOVA (****, p < 0.0001; ***, p < 0.001).

treated for four days instead of five to facilitate LJ removal; T3-induced changes to the overall head structure make removing the LJ more difficult after five days of treatment. Proliferative cells were counted from blinded images over the area of MC-IR (Fig 3A). Fig 3B–3F show

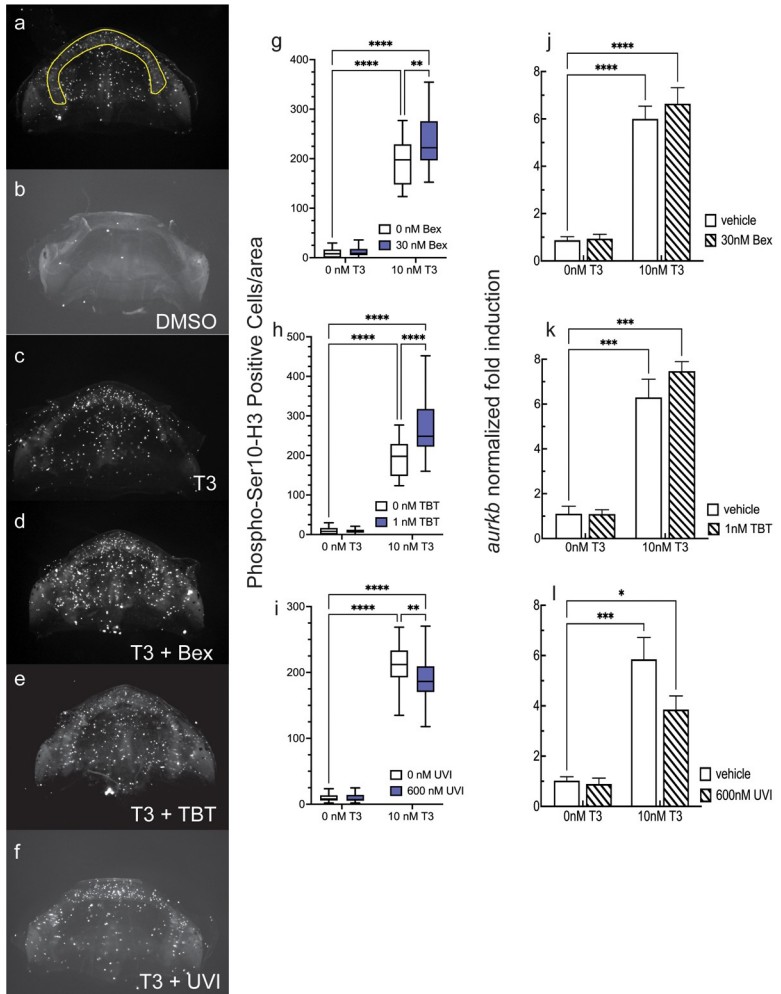

**Fig 3. RXR agonists potentiate T3 action on cellular proliferation in the LJ of 1wk-PF tadpoles.** a: MC-IR region used for quantitation of proliferation. b-f: Representative photos of the effects of different treatments on proliferation using phopho-Ser10-H3 reactivity. g-i: Quantification of proliferation in the presence and absence of T3 and RXR ligands normalized to the area counted. Boxes and statistics are as in Fig 1 (n = 20–30 jaws from 2–3 clutches). j-l: RXR ligands do not significantly affect the T3-induced expression of aurora kinase B mRNA (*aurkb*). Bars represent the mean of 3–6 independent clutches, and statistics show results from Sidak's multiple comparison test in conjunction with 2-way ANOVA (****, p < 0.0001; ***, p < 0.001; **, p < 0.01; *, p < 0.05).

representative photos of different ligand treatment combinations from which proliferative cells were counted and normalized to the area counted. For quantification, each combination of T3 and RXR ligand were assayed with two independent clutches, and for each clutch, RXR ligand effects were significant. Fig 3G–3I shows the two clutches combined for each group. Vehicle-treated LJs had few proliferative cells (Fig 3G–3I). In contrast, treatment with 10 nM T3 increased the number of mitotic cells at least 15-fold for each treatment group. Co-treatment of either 30 nM Bex (Fig 3G) or 1 nM TBT (Fig 3H) RXR agonists with T3 resulted in a significant increase in the number of proliferating cells in the MC-IR cartilage. Since the RXR agonists induced a significant increase in proliferative cells, we expected that co-treatment of the RXR antagonist UVI with T3 would result in a decrease in proliferative cells. Fig 3I shows that UVI significantly inhibited cellular proliferation in the MC-IR. Aurora kinase B (*aurkb*) is the kinase that performs the phosphorylation of Ser10 of H3. T3 induced *aurkb* mRNA expression (Fig 3J–3L). However, neither

Bex (Fig 3J) nor TBT (Fig 3K) significantly increased that induction, and UVI inhibition of *aurkb* was not significantly different from T3-alone (p = 0.081) (Fig 3L). These results suggest that a significant increase in *aurkb* message levels are not necessary for producing the increase in H3-phosphorylation-positive cells seen in T3-RXR agonist co-treatment, and that RXR antagonist inhibition of proliferation doesn't require a significant decrease in *aurkb* expression. Taken together, the data suggest that proliferation is either being controlled by a different mechanism than simply altering *aurkb* expression, or that significant changes in *aurkb* message are not required for significant changes in Aurkb kinase activity.

## RXR agonist potentiation of gene expression is gene specific

Our previous work examining the role of RXR ligands to perturb T3-mediated gene expression in the tails of 1-wk-PF tadpoles after a 48-hour induction, showed that the *bona fide* TR target gene for TRβ, *thrb*, was modestly, but significantly, potentiated by the RXR agonists and inhibited by the antagonist when assayed at the transcriptomic level using Tag-Seq. However, over a time course assayed by RT-qPCR, the same two-day time point showed no significant potentiation and inhibition by the agonists and antagonist, respectively [46]. Using RT-qPCR to assess *thrb* expression in the LJ after two days of treatment, we found significant activation by T3 (white bars in Fig 4A), but neither Bex nor TBT potentiated that induction (slashed bars in Fig 4A, Bex, TBT). UVI also did not inhibit the T3 induction (slashed bar in Fig 4A, UVI). TH-bZIP is a transcription factor that is one of the most strongly TH-induced genes during metamorphosis. It is encoded by the *thibz* gene, and it is another TR direct target gene, having at least two TREs in the promoter region [62]. In the LJ, T3 strongly induced *thibz* expression (Fig 4B, white bars), but the RXR agonists did not potentiate the signal (Fig 4B, slashed bars, Bex, TBT). However, UVI did significantly reduce the T3 induction of *thibz* (Fig 4B, slashed bar, UVI). In the tail, we found the same outcome: the RXR agonists did not affect *thibz* expression, but the RXR antagonist significantly did [43, 46]. These results strongly suggest that the RXR agonists and antagonist are not always operating reciprocally.

During metamorphosis, matrix metalloprotease activity is essential for both tissue resorption and tissue remodeling. We and others have shown the importance of stromelysin-3 (*mmp11*) and collagenase-3 (*mmp13l*) expression [63–65]. We found in the LJ that *mmp11* was strongly activated by T3 (white bars in Fig 4C). Co-treatment with 30 nM Bex increased *mmp11* expression, although this result did not reach statistical significance (p = 0.068), using the maximal number of biological replicates (n = 6) recommended for pooled, outbred animal tissues. However, co-treatment of T3 with 1 nM TBT did result in significantly potentiation of *mmp11* expression (n = 3, p = 0.0004). Furthermore, UVI inhibited the T3 induction of *mmp11* significantly (n = 4, p = 0.001). In the experiments using Bex, even though T3 on its own activated the *mmp13l* gene 20.8-fold (S.E.M. = 4.95) (white bars in Fig 4D), this activation did not reach statistical significance (p = 0.0589), like it does in the tail. However, co-treatment with Bex increased *mmp13l* activation to significance (p < 0.0001) compared to both vehicle and T3-only treatments (slashed bar in Fig 4D, Bex). This situation held true for co-treating T3 with TBT (slashed bar in Fig 4D, TBT): T3-alone activation of *mmp13l* was not significant (p = 0.65), while T3 + TBT treatment was significantly potentiated (p < 0.0001) compared to both vehicle and T3-alone (slashed bar in Fig 4D, TBT). In contrast, in the experiments using UVI, the 7.3-fold activation by T3-alone did reach statistical significance (p = 0.0008, n = 4), but UVI did not significantly inhibit T3 induction of the gene (p = 0.14).

Runx2 is a transcription factor that is required for the transition from proliferating chondrocytes to hypertrophic chondrocytes in the maturation of cartilage for the development of a

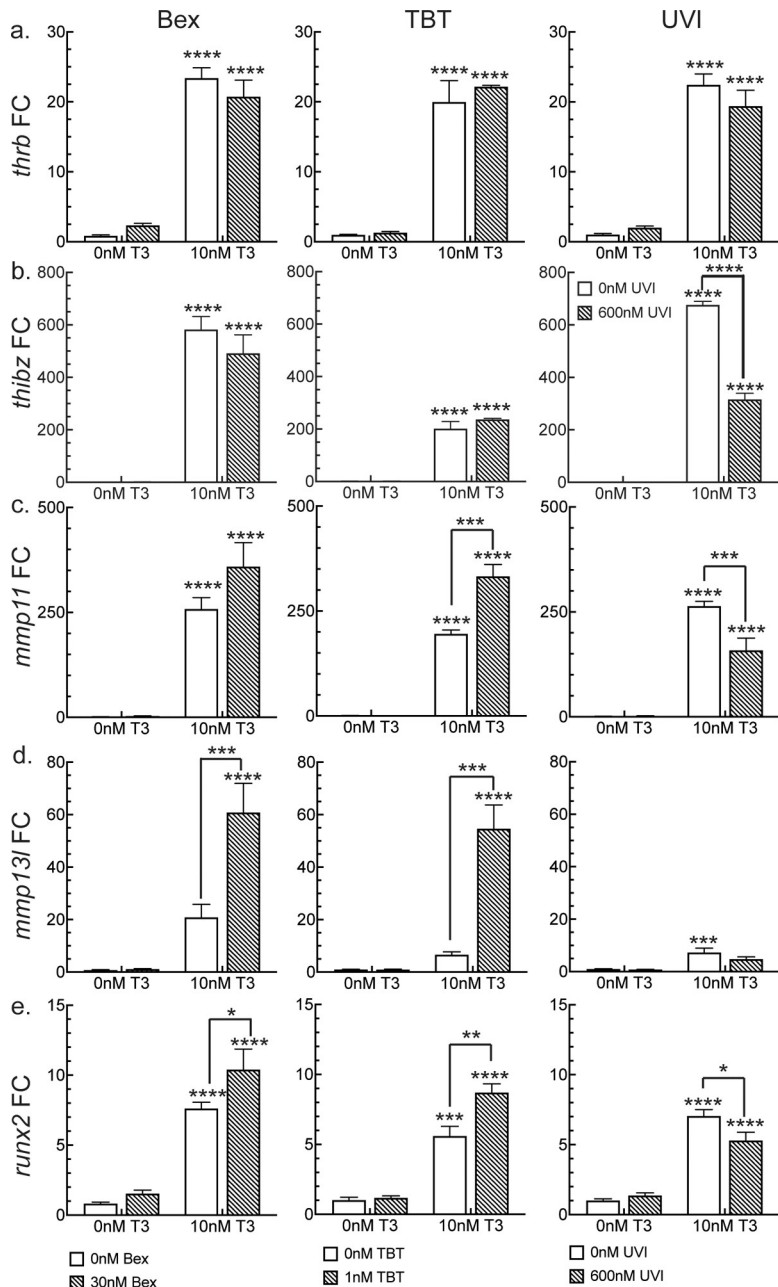

**Fig 4. RXR ligands have gene-specific effects on T3-induced differential gene expression.** Left column: The effect of RXR agonist Bex on T3-induced gene expression. Middle column: The effect of environmental RXR agonist TBT on T3-induced genes. Right column: The effect of RXR antagonist UVI on T3-induced genes. Striped bars indicate the presence of the RXR ligand, and white bars show induction in the absence of the RXR ligand. Statistics show results from Sidak's multiple comparison test in conjunction with 2-way ANOVA (****, $p < 0.0001$; ***, $p < 0.001$; **, $p < 0.01$; *, $p < 0.05$).

bony skeleton [66–68]. In non-amniote animals like fish and amphibia, it is required earlier for rostral cartilage formation [69, 70]. Due to the extensive changes to jaw cartilage during metamorphosis, we investigated whether T3 regulated its expression. In the LJ, T3 induced expression of *runx2* approximately 7-fold (white bars in Fig 4E), and this induction was

significantly potentiated through co-treatment of either Bex or TBT with the T3 (slashed bars in Fig 4E, Bex, TBT). In addition, UVI co-treatment significantly inhibited *runx2* induction by T3 (slashed bar in Fig 4E, UVI).

## RXR agonists potentiate T3-action in TH-competent (NF 54) tadpoles

While 1wk-PF tadpoles are considered only partially competent to respond to THs, tadpoles at NF 54 (approximately 26 days PF) are considered fully competent to respond to THs and to be entering metamorphosis [21, 45, 63]. We raised tadpoles to NF 54, using hind limb development to determine the developmental stage [61], and then treated them with 10 nM T3 in the presence and absence of 30 nM Bex to investigate whether the RXR agonist could still potentiate the action of T3 in a fully competent tadpole. Tadpoles were treated for three days with compounds (a longer treatment time was not possible due to the extreme gill resorption in T3 plus Bex animals), and then we measured the LJ angle. Fig 5A (white boxes) shows that T3-alone caused a small but significant decrease in the lower jaw angle. As in NF 48 tadpoles, Bex-only treatment had no effect on the lower jaw morphology—tadpoles were indistinguishable from vehicle-treated. Bex co-treatment with T3 significantly potentiated the decrease in the LJ angle at this later stage of growth (Fig 5A), suggesting that the ability to increase the competence for T3 in the lower jaw was still possible, even for these presumed fully competent animals.

Previously we developed a transgenic line of *X. laevis* frogs that express firefly luciferase (Luc) under the regulation of the *X. laevis thibz* TH response elements (TREs) [43, 44], where Luc activation follows morphological changes in terms of response to T3 and RXR ligands. The single-copy reporter construct uses only the TRE elements from the *thibz* gene just upstream of a minimal promoter from the MMTV (mouse mammary tumor virus) long terminal repeat [71], in which the response elements for the glucocorticoid receptor were replaced with the TREs from the *thibz* gene. In NF 48 tadpoles, assaying the entire head for Luc activation is required in order to generate a signal robust enough for statistics. At NF 54 we are able to analyze individual tissues, so we treated NF 54 tadpoles for 2 days with 10 nM T3 in the presence and absence of 2 nM TBT, and then we excised the lower jaws as we did for gene expression analysis. Luc activity was determined in the LJ samples and was normalized to the protein concentration of each sample. We assayed two clutches independently using two different TRE-Luc-bearing F2 male frogs to generate embryos with two different wild-type female frogs. TRE-Luc F2 males, even though they arise from the same founder female, display different levels of Luc activation by T3 that are nonetheless consistent within a clutch. Fig 5B shows the results of both clutches individually, showing the different levels of T3 activation between the two clutches. For clutch 2, a TBT-only treatment was also included and showed no activation. Using a 2-way ANOVA analysis of the combined data from both clutches where treatment and clutch were covariates, clutch was a significant source of variance (p = 0.0005), as was treatment (p < 0.0001). Using a Tukey multiple comparisons test post hoc on the combined clutch data, TBT significantly potentiated the T3 activation of the Luc reporter (p = 0.0092). This result indicates that the RXR agonists at this high-TH-competence stage could further increase the competence of LJ tissue for T3 at the beginning of natural metamorphosis.

## Discussion

In this report we have expanded upon our earlier findings concerning the ability of RXR agonists to function as a competence factor for TH signaling during vertebrate development [43, 44, 46]. The poor biological outcomes that arise from insufficient or

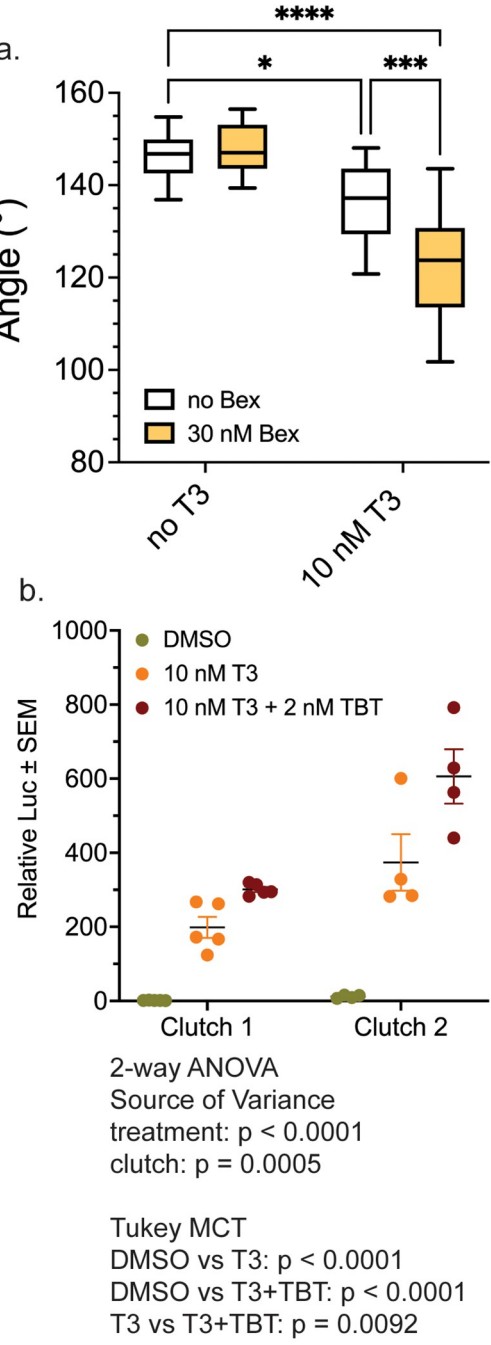

**Fig 5. RXR agonists potentiate T3 action in the LJ in pro-metamorphic NF 54 tadpoles.** a. Bex potentiates the T3-induced decrease in the LJ angle in NF 54 tadpoles treated for three days. Boxes and statistics are as in Fig 1 (n = 14 jaws from 3 clutches). Statistics show results from Sidak's multiple comparison test in conjunction with 2-way ANOVA (****, p < 0.0001; ***, p < 0.001; *, p < 0.05). b. TBT potentiates T3-inducible, integrated luciferase reporter expression in the LJ of NF 54 tadpoles.

inappropriate TH during development have demonstrated the need for assessing the ability of man-made chemicals present in the environment to aberrantly accelerate or inhibit those signaling pathways.

In order to look at TH disruption *in vivo* and during development, we have used amphibian metamorphosis of the African clawed frog, *Xenopus laevis.* Metamorphosis performs two reciprocal functions: 1) development of adult tissues and organs required for life as a frog, and 2) removal of larval tissues no longer needed by the adult frog. Limb formation and growth is an example of development of new tissues, and jaw development is an example of remodeling that must occur for the herbivorous tadpole to become a carnivorous frog. The other side of the metamorphic coin involves the resorption of larval tissues that are no longer required in the frog, such as gills and the tail. Naturally, removal of larval tissues must occur after the adult tissues have developed and become functional. For example, tail resorption is the last step in metamorphosis because it must occur after limb development is complete and the limbs are functional for locomotion. Under natural development, it takes approximately two months to go from a fertilized egg through a larval tadpole to a juvenile frog, with the metamorphic transition from tadpole to frog taking approximately 4.5 weeks under ideal conditions [13, 18, 21, 61].

Our studies here employed a precocious metamorphosis assay, to determine whether a disruptor of TH signaling, which we have previously described disrupting larval tissue resorption phenotypes [43, 46], can also disrupt a larval-to-adult remodeling function, namely, cartilage development in the LJ. By using 1wk-PF tadpoles, we examined tadpoles where the larval jaw cartilages were formed, and the tadpoles had entered a period of isometric growth [51, 52]. This allowed us to focus on the effects of RXR ligands on T3-mediated metamorphic phenotypes as RAR-RXR-mediated specification and differentiation of the cartilage was complete [50]. In this assay system, T3 and the potential disrupting chemicals are taken up by the tadpole through administration in the rearing water. Although the LJ of the 1wk-PF tadpole is not able to support completely normal metamorphic changes to the LJ [53], we found that the LJ can respond to T3 administration with reproducible morphological and molecular readouts.

Previously, we reported that both the pharmaceutical RXR agonist Bex and the environmental RXR agonist TBT disrupted TH signaling in 1wk-PF tadpoles by significantly potentiating the ability of T3 to drive gill and tail resorption. Furthermore, the RXR antagonist UVI abrogated T3 action. Bex and TBT functioned identically in a global transcriptomic analysis of T3 signaling in the tail [46], indicating that TBT was functioning as a bona fide RXR agonist [39, 40, 72–76]. Here, we show that the RXR agonists potentiate T3 action in the LJ by accelerating the rate of change and by increasing the potency of each T3 dose. As in the tail, TBT and Bex behaved nearly identically in the LJ independent of the experimental readout. In addition, the RXR antagonist abrogated the morphological changes induced by T3. We also measured the ability of the agonists and antagonist to disrupt T3-induced cellular proliferation. TBT and Bex both significantly potentiated proliferation, and UVI inhibited proliferation. However, when examining differential gene expression profiles, the agonists and antagonist did not always give reciprocal results. For example, T3 induction of the *thibz* gene was unaffected by the RXR agonists, but was significantly inhibited by the antagonist. This was also seen in tail expression of *thibz* [46]. These data show that the RXR agonists and the antagonist do not always behave in a reciprocal manner at all molecular or cellular targets; therefore, more than one molecular mechanism may be in play.

The T3-induced proliferation did not result in an increase in length of the MC, even when proliferation was potentiated by the RXR agonists. This indicates that the decrease in the LJ angle accompanying T3 exposure did not arise from growth of the MC. A better fit to the morphology patterns observed with RXR agonists and antagonist modulation of the LJ T3 response is the expression patterns of the matrix metalloproteases we tested. Both *mmp11* and *mmp13l* expression levels were potentiated by RXR agonists and inhibited by UVI. The Alcian blue staining of the cranial cartilages did show a metamorphic pattern in that the LJ cartilage was

remodeled, including apparent fusion of the IR with the MC, and the cartilage of the BA (gills) was resorbed.

The transcription factor *runx2*, which in mice is required for formation of ossified bones [77], was also significantly activated by T3 exposure, and that activation was potentiated by the RXR agonists and inhibited by UVI. In *Xenopus* and zebrafish, *runx2* is required earlier for cranial cartilage formation [69, 70]. We believe this is the first example of T3 activating *runx2* expression. In human thyroid cancer and breast cancer cells, TRβ suppressed the expression of *runx2* in the presence of TH, acting as a tumor suppressor [78, 79].

An advantage of using 1wk-PF tadpoles for characterizing disruptors of TH signaling is the size uniformity of the tadpoles. We normally don't have to normalize to the vehicle-treated control in each clutch, as we didn't in Fig 1. However, as the tadpoles age, this size uniformity disappears, making morphological measurements more intrusive, as the animals must be housed separately and anesthetized and photographed before treatment for individual comparisons to after treatment changes. An advantage of assaying the LJ angle, is that it does not scale with tadpole head size; therefore, tadpoles can be group housed and measured only after fixation at the end of treatment. This provides a facile assay for TH disruption over developmental time, which in the case of RXR ligands, as they affect TH competence, could change as the animal develops and intrinsically increases in TH competence.

That said, we also chose NF 54 to assess whether the RXR agonists could still potentiate T3 action in the LJ because that is when plasma T3 is first detectable, and therefore, NF 54 is often considered the dividing line between premetamorphic and metamorphic tadpoles [21]. However, NF 54 is nearly three weeks before metamorphic morphological changes in the jaw become apparent at NF 59 [53], and it is approximately two weeks before exogenous T3 leads to normal metamorphic development in the LJ. Therefore, TH competence in the LJ may still not be complete at NF 54 so that the cartilages can continue to develop in their normal T3-independent fashion until they are in the form that can remodel appropriately to an adult jaw. In addition, we found that TBT was able to potentiate the T3-induction of the TRE-driven luciferase reporter in NF 54 tadpoles at the threshold of metamorphosis. When we assayed whole heads for both gene expression and Luc activity in 1-wk-PF tadpoles, the T3-induction of both the gene and the reporter were potentiated by TBT [43]; in tails we found that the agonists did not potentiate the *thibz* gene [46], which is what we found here in the LJ. We do not know where in the genome the single copy reporter has integrated, and it is formally possible that surrounding genomic context or differences between the endogenous promoter and the reporter promoter outside of the TREs may be driving the different outcomes. Most importantly, we find that the activity of the TRE-Luc reporter most consistently follows the morphological phenotypes in terms of responding to T3 and the RXR agonists and, therefore, is a good surrogate for gene expression data. From our data at NF 54 we can conclude that, as prometamorphosis proceeds, the animal may be vulnerable to inappropriate RXR ligand activity from the environment. Ordinarily, endogenous retinoids can be controlled by the P450 retinoid-degrading enzymes [80, 81], yet organotins, or other as yet unknown chemicals in the environment that activate RXR, evade this buffer, and, therefore, still pose a unique and challenging problem for the exquisitely timed process of metamorphosis.

## Supporting information

**S1 Data. This file contains the underlying data for all figures in the manuscript.**
(XLSX)

## Acknowledgments

The authors would like to acknowledge Professor Bruce Draper for sharing his microscope with color camera with us and Dr. Michael L. Goodson for blinding the images for counting proliferative cells.

## Author Contributions

**Conceptualization:** Brenda J. Mengeling, J. David Furlow.

**Data curation:** Brenda J. Mengeling, Lara F. Vetter.

**Formal analysis:** Brenda J. Mengeling, Lara F. Vetter.

**Funding acquisition:** Brenda J. Mengeling, J. David Furlow.

**Investigation:** Brenda J. Mengeling.

**Methodology:** Brenda J. Mengeling, Lara F. Vetter.

**Project administration:** Brenda J. Mengeling.

**Supervision:** Brenda J. Mengeling, J. David Furlow.

**Validation:** Brenda J. Mengeling.

**Visualization:** Brenda J. Mengeling.

**Writing – original draft:** Brenda J. Mengeling.

**Writing – review & editing:** Brenda J. Mengeling, Lara F. Vetter, J. David Furlow.

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
