## [Decision Letter · Decision Letter 0]

5 Oct 2021

PONE-D-21-28036Retinoid-X Receptor Agonists Increase Thyroid Hormone Competence in Lower Jaw Remodeling of Pre-Metamorphic Xenopus laevis tadpolesPLOS ONE

Dear Dr. Mengeling,

Thank you for submitting your manuscript to PLOS ONE. After careful consideration, we feel that it has merit but does not fully meet PLOS ONE’s publication criteria as it currently stands. Therefore, we invite you to submit a revised version of the manuscript that addresses the points raised during the review process. Aside from addressing the specific comments of the two reviewers, I found reviewer #2's comments particularly compelling.  There is a need for major revision, more appropriate recognition of the body of literature related to the topic, and what appears to be a need for extensive revision and new levels of analysis and perhaps experimental observations.  Whether the necessary changes can be made in a timely manner is not clear to me, and the authors might consider withdrawing the  manuscript to provide time for a more extensive reworking. Please submit your revised manuscript by Nov 19 2021 11:59PM. If you will need more time than this to complete your revisions, please reply to this message or contact the journal office at plosone@plos.org. Please include the following items when submitting your revised manuscript:A rebuttal letter that responds to each point raised by the academic editor and reviewer(s). You should upload this letter as a separate file labeled 'Response to Reviewers'.A marked-up copy of your manuscript that highlights changes made to the original version. You should upload this as a separate file labeled 'Revised Manuscript with Track Changes'.An unmarked version of your revised paper without tracked changes. You should upload this as a separate file labeled 'Manuscript'.

We look forward to receiving your revised manuscript.

Kind regards,

Michael Klymkowsky, Ph.D.

Academic Editor

PLOS ONE

Journal Requirements:

Reviewers' comments:

Reviewer's Responses to Questions

**Comments to the Author**

1. Is the manuscript technically sound, and do the data support the conclusions?

Reviewer #1: Yes

Reviewer #2: Partly

2. Has the statistical analysis been performed appropriately and rigorously? 

Reviewer #1: Yes

Reviewer #2: Yes

3. Have the authors made all data underlying the findings in their manuscript fully available?

Reviewer #1: Yes

Reviewer #2: Yes

4. Is the manuscript presented in an intelligible fashion and written in standard English?

Reviewer #1: Yes

Reviewer #2: Yes

5. Review Comments to the Author

Reviewer #1: Thyroid hormone (TH) signaling plays important roles during vertebrate development and it’s prone to interference from endocrine disruptors. The manuscript entitled “Retinoid-X Receptor Agonists Increase Thyroid Hormone Competence in Lower Jaw Remodeling of Pre-Metamorphic Xenopus laevis tadpoles” reports two retinoid-X receptor agonists, pharmacologic bexarotene (Bex) and environmental tributyltin (TBT), potentiated TH-induced responses in both one-week-old tadpoles that had limited TH-competence and prometamorphic tadpoles at stage NF54 that had full TH-competence. The authors used TH-induced lower jaw remodeling in Xenopus laevis as a model to investigate if Bex and TBT could potentiate TH-induced response in morphology, cellular proliferation and gene expression, respectively, and found that both Bex and TBT increased TH-induced changes of lower jaw angling, cellular proliferation in Meckel’s cartilage, and certain gene expression such as mmp11, mmp13, and runx2, etc. in one-week-old tadpoles. Consistent with these observations, they also showed that UVI 3003, a retinoid-X receptor antagonist, produced opposite phenotypes morphologically and at gene expression level. These phenotypes were partially reproduced in prometamorphic tadpoles at stage NF54. They concluded that the retinoid-X receptor agonists could potentiate TH-induced tissue remodeling in Xenopus laevis, a natural TH-dependent phenomena during frog metamorphosis, though the retinoid-X receptor agonists themselves alone didn’t induce significant morphological or molecular gene expression changes. The manuscript was well-organized and well-written and its publication would benefit readers in the related research field. Below are some issues for consideration.

1. The data showed that both Bex and TBT increased TH-induced cell proliferation in Meckel’s cartilage but UVI didn’t cause change in cell proliferation, though it slightly decreased aurkb expression in the Meckel’s cartilage tissue (Fig. 2). Did the author normalize the counts of proliferating cells to anything, such as the area of the Meckel’s cartilage tissues or total cells in the tissues? Apparently, UVI inhibited the TH-induced lower jaw remodeling, therefore the head of tadpoles treated with TH and UVI looked more like the control tadpoles treated with vehicle only, which had larger heads (Fig. 1). Did they also have larger Meckel’s cartilage tissues? If so, normalization is necessary.

2. TH-induced endogenous thibz expression was not significantly increased from either Bex or TBT treatment in combination with TH (Fig.3b) in one-week-old tadpoles, however, the authors evaluated transgenic firefly luciferase gene expression (Luc) under the control of a thibz promoter and exhibited that TBT increased TH-induced Luc activities in lower jaws of NF54 tadpoles (Fig. 4b). Both the gene expression data and the luciferase activity data were generated from lower jaws, it would be useful to explain the discrepancy of the data? Was the endogenous thibz expression also enhanced under the same treatment in the transgenic animals? What was the rationale for performing the transgenic studies here?

3. In Fig. 2, the combinatory Bex or TBT treatment increased TH-induced cellular proliferation in Meckel’s cartilage, but UVI didn’t caused changes in TH-induced cellular proliferation, assayed by immunostaining of phosphorylated Ser10 of H3 (Fig. g-i). Interestingly, the expression of aurkb gene, which encodes the Aurora kinase B responsible for phosphorylating Ser10 of H3, didn’t change from such combinatory Bex or TBT treatment with TH. Any explanation for the increase in phosphorylated Ser10 of H3 without any change in aurkb? Is the Aurora kinase B the only kinase for phosphorylating Ser10 of H3?

4. Some typos to be corrected: “Sequences for the primers…..given S1 Table.” (page 9, line 198) to be “Sequences for the primers…..given in S1 Table.”; “Interestingly, RXR agonist s and …..” (page 18, line 430) to be “Interestingly, RXR agonists and …..”; “Meckels cartilage” (page 6, line 127, and other places) to be “Meckel’s cartilage”.

Reviewer #2: In this paper Dr Mengeling and co-authors, analyze the effects of treating NF48 or NF54 X. laevis tadpoles with T3 in the presence or not of RXR agonists (bexarotene or tributylin) or agonists UVI3003. They focus specifically on lower jaw morphology and on the effects on cell proliferation and on the expression of selected genes on lower jaw. They conclude that RXR agonists increase TH competence in lower jaw remodeling of X.l. tadpoles.

The conclusion reached of the paper is not innovative, they showed essentially the same conclusion in their 2018 paper in ‘Endocrinology’ “: RXR Ligands Modulate Thyroid Hormone Signaling Competence in Young Xenopus laevis Tadpoles.

Here they focus more specifically on what they call the “angle of Meckels cartilage” which they measure on pictures of whole tadpoles heads. This measure is very superficial, the authors should have analyzed, at a minimum, tadpole cartilage skeletons and described the effects on all skeletal components as, for example, in : Baltzinger M, et al. Dev Dyn. 2005 Dec;234(4):858-67.or Vieux-Rochas M, et al. Birth Defects Res B Dev Reprod Toxicol. 2010. A description of the phenotype is needed e.g. is there any unilateral or bilateral fusion between the Quadrate, the Ceratohyal (Ch) and/or Meclels Cartilage? Can you relate this to specific effects described in the (vast) literature on craniofacial development? Do different treatments result in different phenotypes? No considerations are made on the mechanisms of of action of TRs or RARs, RXR…on craniofacial morphogenesis, on endodermal or epidermal signals to neural crest cells, on CNCCs migration etc . Actually, no considerations at all is made on the potential mechanism, all the vast literature on receptor involvement in the mechanisms of craniofacial morphogenesis is completely ignored.

As this paper does not really provide any novel information and considering the superficiality of the analysis and discussion I am not suggesting the paper to be published.

6. PLOS authors have the option to publish the peer review history of their article (what does this mean?). If published, this will include your full peer review and any attached files.

Reviewer #1: No

Reviewer #2: No

---

## [Author Response · Author response to Decision Letter 0]

14 Mar 2022

Responses to Reviewer 1:

1. Normalization to the size of the cartilage area counted was a very good idea for our proliferation data (previously Figure 2, now Figure 3). We have done this, and our revised results now show significant inhibition of T3-induced proliferation by the RXR antagonist UVI.

2. We have provided an explanation of the differences between the endogenous thibz gene promoter and the promoter in the TRE-Luc reporter in both the results (lines 462-466) and the discussion (lines 576-590). We also explain in the discussion our rationale for using the Luc reporter as it performs as an excellent surrogate to general gene expression changes with treatment (i.e., potentiation or inhibition of T3-action by RXR agonists or antagonist, respectively), and its activity follows morphological changes associated with the treatments.

3. We discuss in the results (lines 363-370) the discrepancy between non-significant changes in aurkb message expression and significant changes in Aurkb kinase activity. The simplest explanation is that Aurkb kinase activity isn’t directly tied to its message level at the level of change we see in the kinase activity. 

4. We corrected all typos that we found in the text.

Response to Reviewer 2:

 We have added a figure (current Figure 2) of Alcian blue cartilage staining of the tadpole head from the ventral side so the reader can visualize the cranial cartilages and how they change in response to T3 and the RXR agonists (lines 219-228, 316-341, 534-536). We measured the length of the Meckel’s cartilage, and we show that its length does not change with treatment. We also validated our LJ angle measurement using dorsal head photos of fixed tadpoles (Figure 1) by measuring the angle formed by the Meckel’s and infrarostral cartilages, showing that both sets of measurement agree with each other. 

 In the introduction (lines 156-167) we briefly explain the timing of the roles of retinoic acids and THs in cranial development, specifying that one advantage of assaying at 1-week post-fertilization is it’s a time when the jaw cartilages are already formed from the action of retinoic acids, but that T3 action of cartilage maturation and bone ossification has not occurred. Therefore, we are perturbing the system at a “quiet” time of isometric growth and not morphological change in the jaw. We also better explain the non-natural changes T3 induces at this stage, and how we are still able to use it as a readout of T3 action and disruption. We did not discuss the specification and migration of the neural crest, because in our system that has already occurred before we start treatments, and therefore, cannot be perturbed by such. To keep the introduction at a reasonable length, we shortened it in other paragraphs, as the tracking shows.

---

## [Decision Letter · Decision Letter 1]

31 Mar 2022

Retinoid-X Receptor Agonists Increase Thyroid Hormone Competence in Lower Jaw Remodeling of Pre-Metamorphic Xenopus laevis tadpoles

PONE-D-21-28036R1

Dear Dr. Mengeling,

We’re pleased to inform you that your manuscript has been judged scientifically suitable for publication and will be formally accepted for publication once it meets all outstanding technical requirements.

Kind regards,

Michael Klymkowsky, Ph.D.

Academic Editor

PLOS ONE

Additional Editor Comments (optional):

Reviewers' comments:

Reviewer's Responses to Questions

**Comments to the Author**

1. If the authors have adequately addressed your comments raised in a previous round of review and you feel that this manuscript is now acceptable for publication, you may indicate that here to bypass the “Comments to the Author” section, enter your conflict of interest statement in the “Confidential to Editor” section, and submit your "Accept" recommendation.

Reviewer #1: All comments have been addressed

Reviewer #2: All comments have been addressed

2. Is the manuscript technically sound, and do the data support the conclusions?

Reviewer #1: Yes

Reviewer #2: Yes

3. Has the statistical analysis been performed appropriately and rigorously? 

Reviewer #1: Yes

Reviewer #2: Yes

4. Have the authors made all data underlying the findings in their manuscript fully available?

Reviewer #1: Yes

Reviewer #2: Yes

5. Is the manuscript presented in an intelligible fashion and written in standard English?

Reviewer #1: Yes

Reviewer #2: Yes

6. Review Comments to the Author

Reviewer #1: Thyroid hormone (TH) signaling plays important roles during vertebrate development and it’s prone to interference from endocrine disruptors. The manuscript entitled “Retinoid-X Receptor Agonists Increase Thyroid Hormone Competence in Lower Jaw Remodeling of Pre-Metamorphic Xenopus laevis tadpoles” reports two retinoid-X receptor agonists, pharmacologic bexarotene (Bex) and environmental tributyltin (TBT), potentiated TH-induced responses in both one-week-old tadpoles that had limited TH-competence and prometamorphic tadpoles at stage NF54 that had full TH-competence. The authors used TH-induced lower jaw remodeling in Xenopus laevis as a model to investigate if Bex and TBT could potentiate TH-induced response in morphology, cellular proliferation and gene expression, respectively, and found that both Bex and TBT increased TH-induced changes of lower jaw angling, cellular proliferation in Meckel’s cartilage, and certain gene expression such as mmp11, mmp13, and runx2, etc. in one-week-old tadpoles. Consistent with these observations, they also showed that UVI 3003, a retinoid-X receptor antagonist, produced opposite phenotypes morphologically and at gene expression level. These phenotypes were partially reproduced in prometamorphic tadpoles at stage NF54. They concluded that the retinoid-X receptor agonists could potentiate TH-induced tissue remodeling in Xenopus laevis, a natural TH-dependent phenomena during frog metamorphosis, though the retinoid-X receptor agonists themselves alone didn’t induce significant morphological or molecular gene expression changes. The manuscript was well-organized and well-written and its publication would benefit readers in the related research field.

The changes and responses are satisfactory in the revised version.

Reviewer #2: The inclusion of the analysis of tadpole cartilage skeletons has greatly improved the quality of the paper.

7. PLOS authors have the option to publish the peer review history of their article (what does this mean?). If published, this will include your full peer review and any attached files.

Reviewer #1: No

Reviewer #2: No

---

## [Editor Report · Acceptance letter]

4 Apr 2022

PONE-D-21-28036R1 

Retinoid-X receptor agonists increase thyroid hormone competence in lower jaw remodeling of pre-metamorphic *Xenopus laevis* tadpoles 

Dear Dr. Mengeling:

I'm pleased to inform you that your manuscript has been deemed suitable for publication in PLOS ONE. Congratulations! Your manuscript is now with our production department. 

Kind regards, 

on behalf of

Dr. Michael Klymkowsky 

Academic Editor

PLOS ONE